# Discontinuation rates and predictors of Medical Cannabis cessation for chronic musculoskeletal pain

Mohammad Khak[1¤], Sina Ramtin[1¤]*, Juliet Chung[1¤], Asif M. Ilyas[1,2,3¤], Ari Greis[1,2,3¤]

1 Rothman Institute Foundation for Opioid Research & Education, Philadelphia, Pennsylvania, United States of America, 2 Rothman Orthopaedic Institute at Thomas Jefferson University, Philadelphia, Pennsylvania, United States of America, 3 Drexel University College of Medicine, Philadelphia, Pennsylvania, United States of America,

¤ Department of Orthopaedic Surgery, Rothman Orthopaedic Institute, Philadelphia, Pennsylvania, United States of America

* sina.ramtin@austin.utexas.edu

## Abstract

Medical cannabis (MC) is increasingly used as a treatment for chronic musculoskeletal pain, but there is limited data on factors influencing its discontinuation. This study aims to investigate the rates of MC discontinuation and explore factors influencing it in patients with chronic musculoskeletal pain. A retrospective analysis was conducted on 78 patients certified for medical cannabis over a 2 year period. Patient demographics, pain origin, and Global Physical Health (GPH) and Global Mental Health (GMH) scores were collected before intervention. Discontinuation rates were measured at three-month and one-year follow-up. Statistical analyses, including Fisher's Exact Test and two-sample t-tests, were performed to assess associations between variables and discontinuation. The overall discontinuation rate of MC use was 57.9% at one year, with 44.7% of patients discontinuing within the first three months. Older age was significantly associated with higher discontinuation rates. Pain origin categories did not significantly differ between those who discontinued and continued MC use, although a higher proportion of patients in the discontinued group reported low back pain. No significant differences were observed in baseline GPH and GMH scores between the two groups. This study demonstrates a high discontinuation rate of MC use in patients for chronic musculoskeletal pain. The absence of significant differences in pain origin or baseline health scores suggests that factors beyond pain location or general health may contribute to the decision to discontinue MC. Further research is needed to explore the long-term effects of MC on pain management and patient outcomes.

**Data availability statement:** The data used in this study were obtained from the Rothman Orthopaedic Cannabis Data Repository (ROCDR), a third-party dataset managed by the Rothman Institute Foundation for Opioid Research & Education. The authors do not have permission to publicly share the full dataset. However, interested researchers may apply for access by contacting the Rothman Institute Foundation via email (Lilly.Caso@rothmanopioid.org). The authors did not receive any special privileges in accessing this dataset.

**Funding:** The author(s) received no specific funding for this work.

**Competing interests:** The authors have declared that no competing interests exist.

## Introduction

Chronic musculoskeletal pain is a prevalent condition that affects the quality of life of millions of individuals worldwide. Despite advancements in pharmacologic and non-pharmacologic therapies, many patients continue to experience persistent pain and disability. As a result, there has been growing interest in alternative treatments, including medical cannabis (MC), for managing chronic pain. The therapeutic application of MC has gained global attention due to its potential to alleviate pain. Cannabis and its derivatives are believed to act on CB1 receptors in the brain, modulating pain signals, and CB2 receptors in the dorsal root ganglion, which influence pain integration in nerve pathways [1]. While MC has shown promise in providing significant relief from chronic pain, often surpassing 50% improvement compared to a placebo [2], its effects on cognition, particularly executive function remain a concern. Short-term memory impairment is commonly associated with MC use, and chronic use can exacerbate cognitive deficits, leading to slower processing and reduced attention. Furthermore, long-term and early-age use of MC has been linked to neurocognitive deficits, with neuroimaging studies showing reduced hippocampal volume and density [3].

Despite the growing acceptance of MC as a therapeutic option for chronic musculoskeletal pain, significant gaps remain in understanding its long-term efficacy. While some patients report significant pain relief, others experience dissatisfaction, intolerance, or prefer more definitive treatments, such as surgery or joint injections [4,5]. These factors contribute to the complexity of integrating MC into routine clinical practice. Furthermore, limited research exists on the reasons for MC discontinuation, which is crucial for optimizing its clinical use.

This study aims to address these gaps by investigating the discontinuation of medical cannabis in patients with chronic musculoskeletal pain. We focus on factors such as age, pain origin, and baseline health status. Understanding the predictors of discontinuation will provide valuable insights into the factors that influence patient decisions regarding MC use, helping guide clinical recommendations and treatment strategies.

## Materials and methods

Patient outcome measures in this study were derived from the Rothman Orthopaedic Cannabis Data Repository (ROCDR), which was established with approval from the Institutional Review Board at Thomas Jefferson University, Philadelphia, Pennsylvania (protocol number 19D.159). Consent was obtained or waived by all participants in this study. Physicians participating in MC certification completed a mandatory four-hour continuing medical education course and sought accreditation from the Pennsylvania Department of Health. The patient certification process involved verifying the presence of one of the 23 state-approved medical conditions, confirming Pennsylvania residency, and reviewing the patient's mental health and pain management history. Eligible individuals underwent a chart review to identify any significant history of substance abuse or any other mental health disorders, with those patients being excluded from the certification process. Additionally, a review of controlled substance

use history was conducted using the Pennsylvania Prescription Drug Monitoring Program. If a patient was currently pre-scribed opioids, their prescribing physician was consulted to discuss the integration of MC into the treatment plan. During the certification visit, patients were provided comprehensive information on the chemical composition of MC, available delivery methods, optimal dosing guidelines, and recommended formulations by the certifying physicians. Upon certifi-cation, patients were issued a medical cannabis identification card by the Pennsylvania Department of Health, enabling them to purchase cannabis at authorized dispensaries within the state.

The medical records of 78 patients diagnosed with chronic musculoskeletal noncancer pain who were certified for MC at the Medical Cannabis Department of the Rothman Orthopaedic Institute between October 2022 and December 2024 were analyzed. These patients, previously described in a prior publication on short-term outcomes [6], were reviewed for a minimum follow-up period of one year to assess MC discontinuation rates. Following their initial MC certification, patients were scheduled to attend follow-up visits at three months and subsequently on an annual basis. To further evaluate the continuity of MC use among these patients, we accessed the Pennsylvania Medical Marijuana Program's patient registry. This registry allows practitioners to verify patients' certification statuses and identify whether they have obtained recerti-fications through other healthcare providers. By cross-referencing our patient cohort with this database, we determined the recertification status of each individual, providing insights into their ongoing engagement with MC treatment across different medical practices.

Patient demographics were obtained through electronic medical records for all patients. At the initial visit, we collected data on pain in specific regions, including low back, neck, shoulder, and foot and ankle pain, as well as the Patient-Reported Outcomes Measurement Information System (PROMIS) global health scale. Due to the variability in the site of musculoskeletal pain, we utilized global pain outcome measures rather than site-specific disability measures. The PROMIS global health scale provided two key scores: Global Physical Health (GPH) and Global Mental Health (GMH) quality of life (QoL) T-scores (available online at https://www.healthmeasures.net/). Statistical analysis for GPH and GMH was conducted using the Shapiro-Wilk test to assess normality. The relationship between explanatory variables and GPH and GMH was tested using T-tests, as the outcome variables were normally distributed. Due to limitations in retrospective chart review, specific details regarding the THC:CBD ratio, delivery method (e.g., vaporized, tincture), and formulation types were not uniformly recorded and could not be analyzed. A p-value of <0.05 was considered statistically significant in the analysis.

## Results

The 3-month and 1-year follow-up data was available for the 76 of 78 patients. By the first 3-month period, 44.7% (34 patients) had discontinued MC. Reasons for discontinuation included dissatisfaction with MC, joint or epidural injections, and surgical intervention. After the 3-month follow-up, an additional 13.2% (10 patients) ceased MC use, resulting in a total discontinuation rate of 57.9% (44 patients) at 1 year. The 1-year study cohort of MC tolerance thus consisted of 32 patients among which 11 patients (14.5%) continued treatment with other physicians outside of our practice (Fig 1).

We compared the characteristics of patients who continued MC use after one year with those who discontinued its use. There was a statistically significant difference in age between the two groups (p = 0.04) (Fig 2). Patients who discontinued MC were older, with a mean age of 71.5 ± 9.8 years compared to those who continued MC, who had a mean age of 64.5 years ± 10.2. There was no significant difference in insurance type between the two groups. Of the 44 patients who dis-continued MC, 47.7% were on Medicare and 52.3% were on HMO/PPO insurance, while 37.5% of those who continued MC were on Medicare and 62.5% were on HMO/PPO. Race distribution did not differ significantly between the groups with 97.7% of patients who discontinued MC being White compared to 90.6% of those who continued MC. Of the patients who discontinued MC, 61.4% were female and 38.6% were male, compared to 65.6% female and 34.4% male in the group that continued MC with no significant difference.

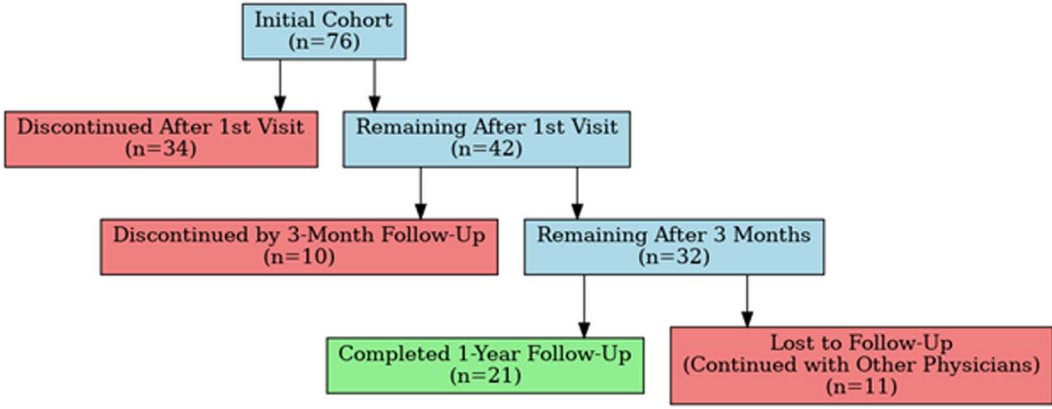

**Fig 1. Patient retention and cannabis use over time.**

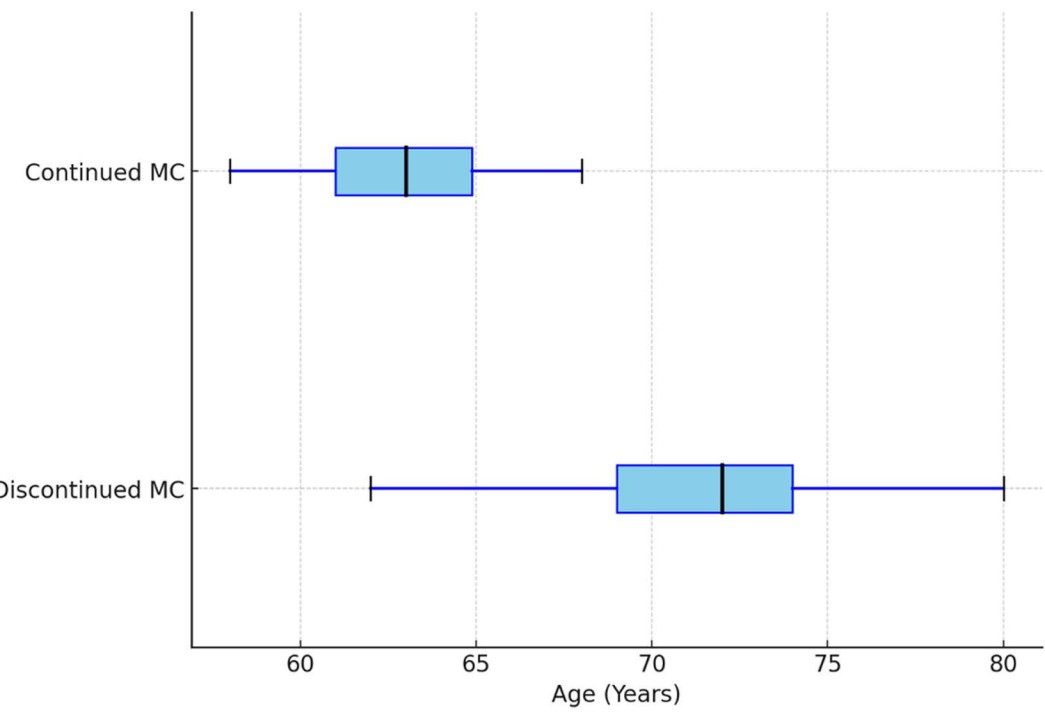

**Fig 2. Comparison of age between patients who discontinued vs. Continued Medical Cannabis (MC) use after 1 Year.**

Regarding health outcomes, there were no significant differences in GPH or GMH scores between the two groups. The mean GPH score for patients who discontinued MC was 43.6±9.8, while those who continued MC had a mean GPH score of 43.9±10.2. Similarly, the mean GMH score for patients who discontinued MC was 46.7±7.5, compared to 47.2±9.3 for those who continued MC (Table 1).

The distribution of pain origin was analyzed between patients who discontinued MC and those who continued its use. The following categories were reported: low back pain, neck pain, Foot/Ankle pain, shoulder pain, and knee pain. We combined all categories associated with low back pain (low back pain only, low back and neck pain, low back and Foot/

**Table 1. Bivariate analysis of discontinued vs. continued Medical Cannabis use.**

| Variable | Continued MC (n = 32) | Discontinued MC (n = 44) | P-value |
|---|---|---|---|
| **Age (Median)** | 64.5 years | 71.5 years | 0.04 |
| **Insurance Type** | | | 0.3 |
| Medicare | 37.5% | 47.7% | |
| HMO/PPO/O | 62.5% | 52.3% | |
| **Race** | | | 0.1 |
| White | 90.6% | 97.7% | |
| Other | 9.4% | 2.3% | |
| **Gender** | | | 0.7 |
| Female | 65.6% | 61.4% | |
| Male | 34.4% | 38.6% | |
| **GPH Score** | 43.9 ± 10.2 | 43.6 ± 9.8 | 0.8 |
| **GMH Score** | 47.2 ± 9.3 | 46.7 ± 7.5 | 0.7 |

Ankle pain, and low back and shoulder pain) and compared the occurrence of low back pain between the two groups. There was no significant difference in the occurrence of low back pain between the 30 patients (68.2%) who discontinued MC and 18 patients (56.3%) who continued MC after 1 year (p = 0.3). For all other pain origin categories, Fisher's Exact Test was applied, and none of the results were statistically significant (all p-values > 0.05) suggesting that the distribution of pain origins between the two groups remained relatively similar across all categories (Fig 3).

## Discussion

This study aimed to investigate the rate of cessation and factors influencing the discontinuation of MC use in patients with chronic musculoskeletal pain. A key finding in this study was the high discontinuation rate, with 57.9% of patients ceasing MC use within one year. This suggests that a large proportion of patients may experience dissatisfaction, intolerance, or prefer alternative treatments such as surgery or joint injections for managing chronic musculoskeletal pain. However, it is important to note that 42.1% of patients either continued MC use or sought recertification, indicating that a subset of patients perceive sustained benefits from MC. These results are consistent with previous studies, which have shown mixed responses to MC treatment in chronic pain patients. While some patients report significant relief, others may not find sufficient therapeutic benefit, leading to early discontinuation [7,8].

By the three-month follow-up, 44.7% had already discontinued MC, reflecting a considerable early drop-off. This early discontinuation could point to initial expectations not being met, potential side effects, or insufficient symptom relief, which are common reasons for discontinuation in medical treatments. The relatively high early discontinuation rate indicates that MC may not provide immediate or sustained relief for all patients and highlights the need for better patient selection and management strategies in the early stages of treatment. To address the high discontinuation rates observed in medical cannabis treatment for chronic pain, recent studies have utilized machine learning models to predict therapy dropout and identify influential factors. Visibelli et al [9] employed a random forest classifier to predict dropout in a cohort of 542 patients undergoing cannabis-based treatment for chronic pain. The study found that high final VAS scores and elevated THC dosages were the most significant predictors of discontinuation, while factors like baseline therapeutic benefits and CBD dosages were associated with improved adherence.

The age of patients was found to be significantly associated with the likelihood of discontinuing MC. Older patients were more likely to cease MC use, which aligns with findings from other studies suggesting that older adults may be more cautious in using alternative therapies like MC due to concerns about long-term effects or a preference for more conventional treatments [10]. Although our study did not collect detailed data on adverse effects, this trend may be partially

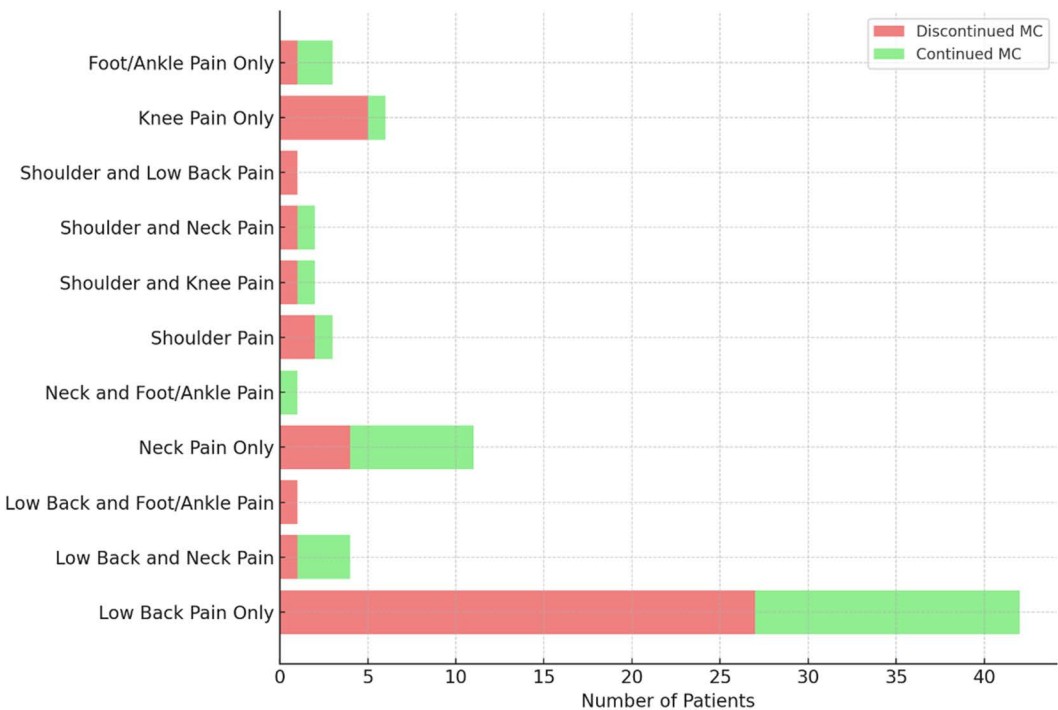

**Fig 3. Pain origin distribution between discontinued and continued Medical Cannabis users.**

explained by age-related concerns such as increased susceptibility to cognitive side effects, dizziness, or drug interactions. One possible explanation for the higher discontinuation rates observed in the elderly population is the presence of more advanced musculoskeletal conditions, such as degenerative changes (radiculopathies, stenosis) or chronic wear and tear (osteoarthritis, tendinopathies), which may not respond well to medical cannabis [10–11]. As patients age, chronic pain often becomes more complex and severe, leading them to seek more definitive treatments such as joint replacements or surgical interventions. Furthermore, the progression of chronic diseases and the presence of comorbidities in older adults may also influence their decision to stop MC, as they may be more inclined to rely on treatments with well-established outcomes while younger individuals tend to continue exploring newer treatment options, including MC. Haug et al. [11] examined cannabis use patterns and motives among medical cannabis dispensary patients of different age groups. They suggest that age-related factors may influence medical cannabis use patterns and the likelihood of discontinuation, which is consistent with our observation that older patients were more likely to discontinue MC treatment.

The origin of pain was also assessed to determine if certain pain conditions influenced MC discontinuation rates over a one-year period. While no significant differences were observed between pain origin categories, we did combine all categories related to low back pain and found that a higher proportion of patients in the Discontinued MC group reported low back pain (68.18%) compared to the Continued MC group (56.25%). Although no statistical significance was reached, this difference suggests that low back pain, as a common and complex chronic condition, may contribute to patients' decisions to discontinue MC. The higher prevalence of low back pain in the Discontinued MC group may reflect the difficulties in managing pain of this nature with MC alone, as patients with chronic low back pain often seek more definitive treatments, such as joint injections or surgical interventions. However, the absence of significant findings points to the multifactorial nature of treatment adherence, where factors beyond pain location, such as psychological, social, and healthcare system-related variables, likely play a more substantial role in the decision to discontinue treatment. Previous studies

have suggested that pain-related factors, including the chronicity and intensity of pain, are often associated with patients' decisions to either persist with or discontinue treatment [12–14].

The distribution of other pain origins, such as neck pain, shoulder pain, and knee pain, did not reveal significant differences between the two groups. These findings suggest that, overall, pain origin does not appear to be a determining factor in whether patients continue or discontinue MC use. This supports the hypothesis that MC may be used as an adjunctive treatment for multiple pain conditions, with its effectiveness varying based on individual patient responses rather than specific pain etiologies. However, this does not negate the importance of individualized treatment plans for patients with complex, multidimensional pain, where factors such as the patient's psychological status, comorbidities, and social support systems are equally important considerations in determining treatment adherence.

While the present study could not capture THC/CBD ratios, specific administration methods, or cognitive effects due to its retrospective design, a recent prospective study of medical cannabis users with chronic musculoskeletal pain has addressed these variables. Another study, which followed patients over a one-year period, reported that most patients used topical formulations, experienced high levels of perceived efficacy, and noted minimal cognitive or motor side effects. In addition, nearly 80% maintained stable usage patterns [15]. These findings offer complementary insights into long-term MC use and may help contextualize the discontinuation patterns observed in this analysis.

Baseline GPH and GMH scores were measured but did not show significant differences between patients who discontinued and those who continued MC use. This suggests that at the time of MC certification, the two groups did not differ substantially in their overall physical or mental health status. However, it is important to recognize that these scores reflect only the baseline health status and do not capture the potential changes in physical and mental health outcomes over time as a result of MC use. Since MC is primarily used to manage chronic pain, it is likely that over time, changes in pain levels and functional status may influence GPH and GMH scores, potentially leading to improved or worsened outcomes that could affect a patient's decision to continue or discontinue treatment [16,17]. Future studies should examine how these scores evolve throughout the course of MC treatment to provide more comprehensive insights into the impact of MC on both physical and mental well-being. This study has several limitations. First, the data on important variables such as THC:CBD ratios, formulation types, and side-effect profiles were not consistently documented and therefore not analyzed. Second, patient-reported outcomes on pain severity, functional improvement, or subjective satisfaction over time were not collected, limiting our ability to understand the underlying reasons for discontinuation (e.g., improvement vs. adverse effects). Third, external factors like cost, dispensary access, and product consistency, known to affect treatment adherence, were not assessed. We also did not evaluate patient expectations or attitudes at baseline, which are known predictors of adherence. Finally, potential confounders such as comorbid cognitive impairment, polypharmacy, or other geriatric syndromes were not included and may have influenced discontinuation, particularly among older adults. The sample was also limited to a specific cohort of patients who sought treatment at a single Medical Cannabis Department, which may not be representative of the broader population of individuals using MC for chronic pain management. Future prospective studies with larger, more diverse populations and more comprehensive data collection on these variables would help to clarify the factors influencing MC use and discontinuation.

## Conclusions

This study highlights the importance of discontinuation rate of MC use in patients with chronic musculoskeletal pain, with over half of patients discontinuing treatment within one year. Age was identified as a key factor influencing discontinuation, with older patients being more likely to stop using MC. While pain origin did not significantly affect discontinuation rates, a higher proportion of patients in the discontinued group reported low back pain, suggesting that the complexity of managing chronic pain conditions with MC alone may contribute to treatment cessation. Additionally, baseline Global Physical Health and Global Mental Health scores did not show significant differences between the two groups, indicating that overall health status at the time of MC certification did not influence the decision to discontinue or continue treatment.

These findings suggest that while MC may offer benefits for some patients, further research is needed to better understand the long-term effects of MC on pain management and patient satisfaction, as well as the factors influencing treatment adherence.

## Author contributions

**Conceptualization:** Mohammad Khak, Sina Ramtin, ASIF M. ILYAS, ARI GREIS.

**Data curation:** Mohammad Khak, Sina Ramtin, JULIET CHUNG.

**Formal analysis:** Mohammad Khak, Sina Ramtin.

**Methodology:** Mohammad Khak, ASIF M. ILYAS.

**Project administration:** Mohammad Khak.

**Software:** Mohammad Khak.

**Supervision:** ASIF M. ILYAS, ARI GREIS.

**Writing – original draft:** Mohammad Khak.

**Writing – review & editing:** Mohammad Khak, Sina Ramtin, ASIF M. ILYAS, ARI GREIS.

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
