## [Decision Letter · Decision Letter 0]

29 May 2025

PONE-D-25-19383
Discontinuation Rates and Predictors of Medical Cannabis Cessation for Chronic Musculoskeletal Pain
PLOS ONE

Dear Dr. Ramtin,

Thank you for submitting your manuscript to PLOS ONE. After careful consideration, we feel that it has merit but does not fully meet PLOS ONE’s publication criteria as it currently stands. Therefore, we invite you to submit a revised version of the manuscript that addresses the points raised during the review process.

**ACADEMIC EDITOR: **

**Please address reviewer comments and resubmit for consideration for publication **

We look forward to receiving your revised manuscript.

Kind regards,

Souparno Mitra, M.D.

Academic Editor

PLOS ONE

Journal Requirements:

3. For studies involving third-party data, we encourage authors to share any data specific to their analyses that they can legally distribute. PLOS recognizes, however, that authors may be using third-party data they do not have the rights to share. When third-party data cannot be publicly shared, authors must provide all information necessary for interested researchers to apply to gain access to the data. (https://journals.plos.org/plosone/s/data-availability#loc-acceptable-data-access-restrictions)

4. Please remove all personal information, ensure that the data shared are in accordance with participant consent, and re-upload a fully anonymized data set.

5. We note you have included a table to which you do not refer in the text of your manuscript. Please ensure that you refer to Table 1 in your text; if accepted, production will need this reference to link the reader to the Table.

Reviewers' comments:

Reviewer's Responses to Questions

**Comments to the Author**

1. Is the manuscript technically sound, and do the data support the conclusions?

Reviewer #1: Partly

Reviewer #2: Yes

2. Has the statistical analysis been performed appropriately and rigorously? 

Reviewer #1: I Don't Know

Reviewer #2: Yes

3. Have the authors made all data underlying the findings in their manuscript fully available?

Reviewer #1: Yes

Reviewer #2: Yes

4. Is the manuscript presented in an intelligible fashion and written in standard English?

Reviewer #1: Yes

Reviewer #2: Yes

5. Review Comments to the Author

Reviewer #1: The study titled “Discontinuation Rates and Predictors of Medical Cannabis Cessation for Chronic Musculoskeletal Pain” addresses a timely and important question on medical cannabis (MC) use in chronic musculoskeletal pain, a condition with significant public health burden. This study uniquely examines discontinuation rates and potential predictors, filling a knowledge gap. Data is collected from a clinical MC program, reflecting real-world usage and discontinuation patterns in a regulated environment. Analysis at 3 months and 1 year provides insight into early attrition versus sustained engagement, which is valuable for guiding clinical expectations and patient counseling.

However some of the limitations have not been addressed or mentioned in the study:

1. Discontinuation may vary by THC/CBD ratio, method of administration (e.g., tincture vs. vaporized), and side-effect profiles. These key variables were not collected or analyzed.

2. Without tracking pain severity or functional status over time, it's unclear whether discontinuation reflects symptom improvement, lack of efficacy, or adverse effects.

3. Patient expectations and subjective satisfaction are strong predictors of treatment adherence, but these were not assessed.

4. Financial barriers and cannabis availability (e.g., dispensary access, product consistency) may influence discontinuation but are not addressed.

5. Other comorbid conditions (e.g., cognitive impairment, polypharmacy) in older adults could influence discontinuation but were not explored.

Please address these limitations in the study.

Reviewer #2: Well written article.

Please explain and further expand on the various complications arising as a result of medical cannabis use which led to discontinuation of use mainly in the older population.

Also an explanation of the types of Medical Cannabis used by the subjects in the study , with mention of the THC:CBD ratio would give a broader understanding of what types of Medical Cannabis were tolerated and which ones were discontinued more.

6. PLOS authors have the option to publish the peer review history of their article (what does this mean?). If published, this will include your full peer review and any attached files.

Reviewer #1: **Yes: **Nikhil Tondehal

Reviewer #2: **Yes: **Arun Prasad

---

## [Author Response · Author response to Decision Letter 1]

8 Jun 2025

Dear Editor-in-Chief of PLOS ONE,

We sincerely thank you for the opportunity to revise and resubmit our manuscript titled:

"Discontinuation Rates and Predictors of Medical Cannabis Cessation for Chronic Musculoskeletal Pain."

We appreciate the thoughtful comments provided by the editorial team and peer reviewers. We have revised the manuscript accordingly and addressed each point as detailed below. All changes have been incorporated into the main text per PLOS ONE guidelines.

Please find our detailed responses below:

Editorial Comments

1. Style and Formatting:

Please ensure that your manuscript meets PLOS ONE's style requirements…

Response:

We have reviewed and revised the manuscript to align with the PLOS ONE formatting style, including the title page, abstract structure, and section headings.

2. Ethics Statement:

Please include your full ethics statement in the ‘Methods’ section…

Response:

We have added the following ethics statement in the “Methods” section:

“Patient outcome measures in this study were derived from the Rothman Orthopaedic Cannabis Data Repository (ROCDR), which was established with approval from the Institutional Review Board at Thomas Jefferson University, Philadelphia, Pennsylvania (protocol number 19D.159). Verbal consent was obtained from all participants prior to data collection.”

3. Data Availability – Third-Party Data:

Please describe the data source, confirm usage permission, disclose any special access, and provide contact info for data access.

Response:

We have included the following statement in the Data Availability section:

“The data used in this study were obtained from the Rothman Orthopaedic Cannabis Data Repository (ROCDR), a third-party dataset managed by the Rothman Institute Foundation for Opioid Research & Education. The authors do not have permission to publicly share the full dataset. However, interested researchers may apply for access by contacting the Rothman Institute Foundation at mohammad.khak@rothmanopioid.org. The authors did not receive any special privileges in accessing this dataset.”

4. Data Anonymization:

Remove personal information and re-upload an anonymized dataset.

Response:

As outlined in our Data Availability Statement (Editorial Comment #3), the dataset used in this study is part of the Rothman Orthopaedic Cannabis Data Repository (ROCDR), a third-party database we do not have legal rights to publicly share. Therefore, no dataset has been uploaded.

5. Missing Table Reference:

Refer to Table 1 in the text.

Response:

We have added a reference to Table 1 in the Results section.

Reviewer Comments

Reviewer #1

We appreciate the reviewer’s valuable feedback. We have added a new paragraph in the Discussion section addressing the following limitations. We also added a reference to a similar recently published article addressing part of these limitations in a prospective study.

Reviewer #2 Comment 1:

Please expand on complications in older patients that may have led to discontinuation.

Response:

We added the following paragraph to the Discussion section:

“Although our study did not collect detailed data on adverse effects, this trend may be partially explained by age-related concerns such as increased susceptibility to cognitive side effects, dizziness, or drug interactions. One possible explanation for the higher discontinuation rates observed in the elderly population is the presence of more advanced musculoskeletal conditions, which may not respond well to medical cannabis.”

Reviewer #2 Comment 2:

Explain types of cannabis used and include THC:CBD ratios.

Response:

We acknowledge this limitation. Unfortunately, detailed data on product formulation and THC:CBD ratios were not consistently available in patient records. We have noted this as a limitation in the Methods and Discussion and highlighted the need for future research that includes specific product characteristics to better understand their impact on treatment adherence.

---

## [Decision Letter · Decision Letter 1]

3 Jul 2025

PONE-D-25-19383R1
Discontinuation Rates and Predictors of Medical Cannabis Cessation for Chronic Musculoskeletal Pain
PLOS ONE

Dear Dr. Ramtin,

Thank you for submitting your manuscript to PLOS ONE. After careful consideration, we feel that it has merit but does not fully meet PLOS ONE’s publication criteria as it currently stands. Therefore, we invite you to submit a revised version of the manuscript that addresses the points raised during the review process.

**Please address reviewer comments and resubmit to reconsider for publication.**

We look forward to receiving your revised manuscript.

Kind regards,

Souparno Mitra, M.D.

Academic Editor

PLOS ONE

**Journal Requirements:**

Reviewers' comments:

Reviewer's Responses to Questions

**Comments to the Author**

1. If the authors have adequately addressed your comments raised in a previous round of review and you feel that this manuscript is now acceptable for publication, you may indicate that here to bypass the “Comments to the Author” section, enter your conflict of interest statement in the “Confidential to Editor” section, and submit your "Accept" recommendation.

Reviewer #2: All comments have been addressed

Reviewer #3: (No Response)

2. Is the manuscript technically sound, and do the data support the conclusions?

Reviewer #2: Partly

Reviewer #3: Yes

3. Has the statistical analysis been performed appropriately and rigorously? 

Reviewer #2: Yes

Reviewer #3: Yes

4. Have the authors made all data underlying the findings in their manuscript fully available?

Reviewer #2: Yes

Reviewer #3: Yes

5. Is the manuscript presented in an intelligible fashion and written in standard English?

Reviewer #2: Yes

Reviewer #3: Yes

6. Review Comments to the Author

**Reviewer #2: **Thank you for addressing the suggested edits for "Discontinuation Rates and Predictors of Medical Cannabis Cessation for Chronic Musculoskeletal Pain."

**Reviewer #3: **“scores were collected at baseline”. Recommend substituting Baseline with before intervention.

“Executive function” includes Working Memory and selective attention

“history of substance abuse or severe mental health disorders”. Substance abuse is mental health disorder when they meet criteria per DSM-V

Include PROMIS 10 questionnaires using 5-point Likert scales. Rating is “limited to the past 7 days” before intervention.

“There was a significant difference in age between the two groups” Statistically significant

Reference for “advanced musculoskeletal conditions which may not respond well to medical cannabis”

“Advanced musculoskeletal conditions in the elderly” such as degenerative changes (radiculopathies, stenosis) or chronic wear and tear(OA, tendinopathies etc)

“That study.” Please reference the study here

“due to its retrospective design”. Does 2-year follow-up after an intervention(MC) make it a prospective rather than a retrospective study?

7. PLOS authors have the option to publish the peer review history of their article (what does this mean?). If published, this will include your full peer review and any attached files.

Reviewer #2: **Yes: **Arun Prasad

Reviewer #3: **Yes: **Anoop Narahari

---

## [Author Response · Author response to Decision Letter 2]

8 Jul 2025

Dear Editor-in-Chief of PLOS ONE,

We sincerely thank you for the opportunity to revise and resubmit our manuscript titled:

"Discontinuation Rates and Predictors of Medical Cannabis Cessation for Chronic Musculoskeletal Pain."

We appreciate the thoughtful comments provided by the editorial team and peer reviewers. We have revised the manuscript accordingly and addressed each point as detailed below. All changes have been incorporated into the main text per PLOS ONE guidelines.

Please find our detailed responses below:

Reviewer#2 Comments:

1. “scores were collected at baseline”. Recommend substituting Baseline with before intervention.

Response:

Corrected. The term “baseline” has been replaced with “before intervention” throughout the manuscript.

2. “Executive function” includes Working Memory and selective attention

Response:

We have revised the manuscript to remove specific mentions of “working memory” and “selective attention” under “executive function” to improve clarity and accuracy.

3. “history of substance abuse or severe mental health disorders”. Substance abuse is a mental health disorder when it meets DSM-V criteria

Response:

Thank you for the clarification. The sentence has been updated to: “significant history of substance abuse or any other mental health disorders” to align with DSM-V definitions.

4. Include PROMIS 10 questionnaires using 5-point Likert scales. Rating is “limited to the past 7 days” before intervention.

Response:

We have added a reference to the questionnaire, which is available online at https://www.healthmeasures.net/.

5. “There was a significant difference in age between the two groups” – Statistically significant

Response:

We have updated the sentence to read: “There was a statistically significant difference in age between the two groups.”

6. Reference for “advanced musculoskeletal conditions which may not respond well to medical cannabis”

Response:

A supporting reference has been added to the manuscript to substantiate this statement.

7. “Advanced musculoskeletal conditions in the elderly” such as degenerative changes (radiculopathies, stenosis) or chronic wear and tear (OA, tendinopathies etc)

Response:

We have incorporated these specific examples—radiculopathies, stenosis, osteoarthritis (OA), and tendinopathies—into the manuscript to provide clarity.

8. “That study.” Please reference the study here

Response:

The study has been referenced as citation [15], and the sentence has been rephrased accordingly.

9. “due to its retrospective design”. Does 2-year follow-up after an intervention (MC) make it a prospective rather than a retrospective study?

Response:

To eliminate confusion, we have removed this sentence from the manuscript.

---

## [Decision Letter · Decision Letter 2]

23 Jul 2025

Discontinuation Rates and Predictors of Medical Cannabis Cessation for Chronic Musculoskeletal Pain

PONE-D-25-19383R2

Dear Dr. Ramtin,

We’re pleased to inform you that your manuscript has been judged scientifically suitable for publication and will be formally accepted for publication once it meets all outstanding technical requirements.

Kind regards,

Souparno Mitra, M.D.

Academic Editor

PLOS ONE

Additional Editor Comments (optional):

Reviewers' comments:

Reviewer's Responses to Questions

**Comments to the Author**

1. If the authors have adequately addressed your comments raised in a previous round of review and you feel that this manuscript is now acceptable for publication, you may indicate that here to bypass the “Comments to the Author” section, enter your conflict of interest statement in the “Confidential to Editor” section, and submit your "Accept" recommendation.

Reviewer #2: All comments have been addressed

Reviewer #3: All comments have been addressed

2. Is the manuscript technically sound, and do the data support the conclusions?

Reviewer #2: Yes

Reviewer #3: Yes

3. Has the statistical analysis been performed appropriately and rigorously? 

Reviewer #2: Yes

Reviewer #3: Yes

4. Have the authors made all data underlying the findings in their manuscript fully available?

Reviewer #2: Yes

Reviewer #3: Yes

5. Is the manuscript presented in an intelligible fashion and written in standard English?

Reviewer #2: Yes

Reviewer #3: Yes

6. Review Comments to the Author

Reviewer #2: Thank you for making the suggested edits. Best of luck on publication of "Discontinuation Rates and Predictors of Medical Cannabis Cessation for Chronic Musculoskeletal Pain.

Reviewer #3: (No Response)

7. PLOS authors have the option to publish the peer review history of their article (what does this mean?). If published, this will include your full peer review and any attached files.

Reviewer #2: **Yes: **Arun Prasad

Reviewer #3: **Yes: **Anoop Narahari

---

## [Editor Report · Acceptance letter]

PONE-D-25-19383R2

PLOS ONE

Dear Dr. Ramtin,

I'm pleased to inform you that your manuscript has been deemed suitable for publication in PLOS ONE. Congratulations! Your manuscript is now being handed over to our production team.

Kind regards,

on behalf of

Dr. Souparno Mitra

Academic Editor

PLOS ONE